# Comprehensive Characterization and Functional Analysis of the Lateral Organ Boundaries Domain Gene Family in Rice: Evolution, Expression, and Stress Response

**DOI:** 10.3390/ijms26093948

**Published:** 2025-04-22

**Authors:** Shang Sun, Jingjing Yi, Peiling Gu, Yongtian Huang, Xin Huang, Hanqing Li, Tingting Fan, Jing Zhao, Ruozhong Wang, Mahmoud Mohamed Gaballah, Langtao Xiao, Haiou Li

**Affiliations:** 1Hunan Provincial Key Laboratory of Phytohormones and Growth Development, Hunan Agricultural University, Changsha 410128, China15574111279@163.com (X.H.); lihanqing2025@163.com (H.L.); ste0622@163.com (T.F.); wangruoz@163.com (R.W.); 2Yuelushan Laboratory, Changsha 410128, China; 3Rice Research and Training Center (RRTC), Field Crops Research Institute, Agricultural Research Center, Sakha, Kafr Elsheikh 33717, Egypt; mahmoudgab@yahoo.com

**Keywords:** rice, LBD gene family, system evolution, gene expression, stress response

## Abstract

In this study, the LBD (Lateral Organ Boundaries Domain) gene family, a group of plant-specific transcription factors critical for plant growth and development as well as metabolic regulation, was comprehensively characterized in rice. We identified 36 LBD genes using multi-source genomic data and systematically classified them into Class I (31 genes) and Class II (5 genes). Analysis of their physicochemical properties revealed significant variations in amino acid length, molecular weight, isoelectric points, and hydropathicity. Motif analysis identified conserved LOB domains and other motifs potentially linked to functional diversity. Cis-acting element analysis indicated the involvement of these genes in various biological processes, including light response, hormone signaling, and stress response. Expression profiling demonstrated tissue-specific expression patterns, with several genes, such as *XM_015770711.2*, *XM_015776632.2*, and *XM_015792766.2*, showing relatively high expression in rice roots, implying their important role in root development. Transcriptome data further supported the involvement of specific genes in responses to phytohormones such as jasmonic acid (JA) and abscisic acid (ABA), as well as environmental stresses like cold and drought. Notably, *XM_015770711.2*, *XM_015776632.2*, and *XM_015772758.2* may contribute to the regulation of rice environmental adaptability by mediating ABA and JA signaling pathways, respectively. In conclusion, this study identified members of the LBD gene family through the screening of two rice gene databases, and performed a comprehensive analysis of their physicochemical properties, evolutionary relationships, and expression profiles under various conditions. These findings provided valuable insights for further functional studies of LBD genes. Moreover, this study provides a foundation for targeting LBD genes to enhance stress resilience (e.g., drought/cold tolerance) and root architecture optimization. The LBD gene family possesses dual values in both stress resistance regulation and developmental optimization. The construction of its multidimensional functional map lays the theoretical and resource foundation for the precise design of high-yield and stress-resistant varieties.

## 1. Introduction

Transcription factors (TFs) are a class of proteins that bind to DNA in a sequence-specific manner. They are commonly found in higher plants which regulate gene expression, playing important regulatory roles in various biological processes. The LBD transcription factor family is a kind of plant-specific transcription factor [1]. It is named for the inclusion of a conserved lateral organ boundary (LOB) domain, which consists of more than 100 amino acid residues. The conserved LOB domain consists of three parts, listed in order from N-terminal to C-terminal: the highly conserved C motif (C-block, CX2CX6CX3C), the Gly-Ala-Ser motif (GAS-block), and the leucine zipper-like block (LX6LX3LX6L). The C motif is essential for DNA binding and contains four cysteine residues, while ‘X’ represents an unconserved residue [2,3], The GAS motif influences DNA binding activity, while the leucine zipper-like block is primarily responsible for protein dimerization. Based on the completeness of the leucine zipper-like block, LBD gene family members are classified into two subfamilies (Class I and Class II): those with a complete leucine zipper-like block are classified as Class I, while those lacking the motif are classified as Class II. As the leucine zipper-like block is related to protein dimerization, Class II LBD genes are unable to form a coiled-coil structure. However, Class II LBD genes are more conserved, and the similarity between different genes is very high. With advances in molecular biology and bioinformatics, the identification and analysis of the LBD gene family have been carried out in several species., such as Arabidopsis thaliana [3], Brassica napus [4], Solanum tuberosum [5], Gossypium [6], Triticum aestivum [7], and other plants [8,9,10].

The LBD gene family plays an important role in multiple processes of plant growth and development as well as metabolic regulation, such as in plant lateral root development [11], anthocyanin synthesis [12], pollen development [13], vascular tissue formation [14], and callus formation [15]. In *Arabidopsis thaliana*, Auxin Response Factor 7(ARF7) and Auxin Response Factor 9(ARF19) can directly bind to the promoter regions of *AtLBDl6* and *AtLBD29* to activate gene expression to control lateral root formation. In the *arf7arf19* double mutant, lateral root formation was completely blocked, while overexpression of *AtLBDl6* or *AtLBD29* partially restored lateral root formation in the mutant [16]. In addition, LBD29 is also involved in auxin-regulated callus formation [17]. AtLBD18 is involved in lateral root initiation. *AtLBD18* can directly activate the transcription of a cell wall relaxation factor encoded by *EXPANSIN4*, which can form homologous or heterologous dimers with AtLBD16 to synergistically act downstream of *ARF7* and *ARF19* [18,19]. *AtLBD33* and *AtLBD18* are involved in lateral root initiation by synergistically regulating the activation of *E2Fa* transcription, which, in *Arabidopsis thaliana*, is a key factor in regulating asymmetric cell division during lateral root initiation. AtLBD18 can directly bind to the LBD motif in the promoter region of *E2Fa* to activate its transcription [20]. In addition, LBD18 can also regulate plant duct differentiation [21]. The LBD gene family also plays an important role in root growth and development in other plants, such as *OsLBD1-8(XM_015780552.2*), the expression of which is regulated by an AUX/IAA-ARF complex, a key regulator of both constitutive aerenchyma and lateral root formation [22]; *TaMOR* (more root in wheat) determines the initiation of wheat canopy roots [23]; *CmLBD1* is responsive to auxin and has been shown to be involved in the formation of adventitious root primordium in *Chrysanthemum*, and its heterologous expression in *Arabidopsis thaliana* increases the number of lateral root formed [24].

*AtLBD6/Asymmetric Leaves2 (AS2) is* involved in the developmental regulation of adaxial–abaxial polarity by repressing the expression *of KNOTTED1-like HOMEOBOX (KNOX) genes in lateral organs* [25]; transgenic rice expressing *OsAS2* exhibited abnormally twisted leaves, lacked auricle, and abnormal leaf structures [26]. *AS2* is also involved in the development of floral organs in *Arabidopsis thaliana*. It works synergistically with *Asymmetric Leaves1* (*AS1*) and *JAGGED* (*JAG*) to control gene expression at the primordium boundary of organs [27]. The synergistic effect of *AtLBD10* and *AtLBD27* plays a key role in pollen development in *Arabidopsis thaliana*. Genetic experiments showed that 10–15% of pollens in the *lbd10* mutant were dysplastic, along with and 70% in the *lbd27* mutant, while in the *lbd10 lbd27* double mutant, all pollens were wrinkled and aborted [28]. *OsIG1*(*Indeterminate Gametophyte1*) down-expression lines through RNAi exhibited pleiotropic phenotypes, such as degenerated palea, open hull and double ovules with abnormal embryo sac development in some florets, which suggested that *OsIG1* plays an important role in the reproductive development of rice [29]. In addition, AtLBD1, AtLBD3, AtLBD4, and AtLBD11 can negatively regulate cytokinin signaling, which maintains the secondary growth of plants [30]. The negative regulation of anthocyanin biosynthesis in *Arabidopsis thaliana* is mediated by AtLBD37, AtLBD38, and AtLBD39. In the absence of nitrogen/nitrate (N/NO_3_^−^), overexpression of any of the three genes could significantly inhibit the expression of the essential regulators of anthocyanin biosynthesis, namely *Anthocyanin Pigment 1(PAP1*) and *Anthocyanin Pigment 2 (PAP2*), as well as the genes involved in the anthocyanin segment of flavonoid synthesis [12]. OsLBD37 and OsLBD38 can delay heading date by down-regulating the expression of the florigen genes *Heading date 3a* (*Hd3a*) and *Rice Flowering Locus T1* (*RFT1*) through the key regulator of *Early heading date 1* (*Ehd1*) [31].

The LBD gene family also plays a significant role in plant secondary growth and resistance in other species. PheLBD29, in *Phyllostachys pubescens*, induces leaf curvature and enhances drought tolerance in transgenic *Arabidopsis thaliana* [32]. In poplar, Dominant-negative suppression of Pta LBD1 via translational fusion with the repressor SRDX domain caused decreased diameter growth and suppressed highly irregular phloem development [33]. Similarly, in *Populus*, overexpression of *PagLBD3* enhanced secondary growth in the stem with a marked increase in the rate of cambial cell differentiation into phloem. Conversely, dominant repression of *PagLBD3* resulted in a reverse phenotype [34]. Meanwhile, the LBD gene family also plays an important role in plant resistance to stress. SlLBD40, which participates in jasmonic acid (JA) signal transduction, acts as a negative regulator of drought tolerance, and knockout of *SlLBD40* can enhance the drought tolerance of tomato and overexpression of *SlLBD40* remarkablely accelerated fruit growth by stimulating mesocarp cell expansion, whereas knockout of *SlLBD40* showed the opposite outcome [35,36]. *LBD20* serves as a susceptibility gene for *Fusarium oxysporum*, seemingly regulating components of JA signaling downstream of *Coronatine Insensitive1* (*COI1*) and *MYC2*, which are crucial for the complete elicitation of *F. oxysporum* and JA-dependent responses [37].

As mentioned above, the LBD gene family has been extensively studied in Arabidopsis and various other plants, but the functions of many members of this family remain unexplored, especially in rice. Rice (*Oryza sativa*) has a long history of cultivation and consumption and is not only one of the important food crops for humans but also an excellent model for studying monocotyledonous plants. Members of homologous gene families often exhibit both functional correlation and uniqueness. Therefore, understanding gene family members requires consideration of both orthologous and paralogous genes. Based on previous studies on the functions of LBD family members, we believe that LBD genes in rice are extensively involved in various aspects of growth and development, including root architecture formation, secondary growth, metabolism, and stress responses. In this study, we aim to identify LBD genes in rice by screening two distinct genomic databases to ensure the accuracy and comprehensiveness of gene identification. We will conduct an in-depth analysis of the physicochemical properties, conserved motifs, cis-acting regulatory elements, phylogenetic relationships, gene collinearity, and expression profiles of LBD family members. By integrating existing functional studies, we seek to validate our hypothesis that the LBD gene family plays a significant role in regulating rice growth, development, and responses to various environmental stresses such as cold and drought. This research will provide important molecular evidence supporting the functional involvement of LBD genes and will lay a solid foundation for future in-depth investigations into their biological roles and potential applications in rice breeding.

## 2. Result

### 2.1. Identification and Physicochemical Property Analysis of the Rice LBD Gene Family Members

BLASTP and HMMER alignments were performed separately to obtain the rice LBD gene family members. Among them, BLASTP can rapidly compare protein sequences, accurately identify homologous proteins with medium to high similarity, and provide reliable statistical evaluations. It is a core tool in protein analysis. HMMER is based on profile Hidden Markov Model (HMM) technology. Through highly efficient algorithms, it enables fast and sensitive detection of distantly related homologs, providing support for genome annotation and large-scale analysis of core databases such as Pfam.

The rice genome data were obtained from NCBI. Protein sequences of 43 *Arabidopsis thaliana* LBD gene family members were derived in the plant transcription factor database as query sequences for local BLASTP alignment, and 45 candidate sequences of rice LBD gene family were obtained. Meanwhile, the Pfam registration number of the typical structure of LBD was identified as PF03195 after query, and its hidden Markov model was downloaded, and 45 rice LBD gene family candidate sequences were obtained by HMMER alignment. We employed ClustalW for multiple sequence alignment and utilized NCBI CDD tool to detect the domains of the candidate sequences and manually remove duplicate sequences in order to screen out valid genes. As a result, 35 LBD genes were obtained from the screening. At the same time, we named the selected genes based on the information in the two databases and their homologous genes in *Arabidopsis thaliana*. Gene names that were repeated in the database were assigned additional numbers based on the descending order of chromosome numbers.

During the follow-up analysis, it was found that some genes might have been overlooked; as such, the genome data of rice were acquired from Rice Genome Annotation Project again and the above procedure was repeated to finally obtain 35 LBD genes. The LBD genes obtained from the two databases were aligned, and it was found that a gene *LOC_Os03g57670.1*, with a complete LOB domain from Rice Genome Annotation Project database, was successfully aligned with the gene *XM_026023997.1* from NCBI database, while *XM_026023997.1* in the NCBI database was eliminated during screening due to its incomplete LOB domain, and the complete genes were subsequently integrated into the NCBI database, and finally 36 LBD genes were acquired (Figure 1).

The results of ExPASy analysis showed that rice LBD gene family members differed in the physicochemical properties of their protein sequences, the number of amino acids ranged from 117 to 456, the molecular weight ranged from 12,500.28 to 48,653.41 Da, the isoelectric point ranged from 4.62 to 9.72, the hydropathicity ranged from −0.88 to 0.125, and that most members are hydrophobic. The predicted subcellular localization analysis of Cell-PLoc 2.0 suggested that all the members were located in the nucleus (Table 1).

Due to the controversial prediction results of subcellular localization from previous studies (e.g., mitochondria, chloroplast thylakoid membrane, and extracellular space) [38], we selected three genes *XM_015770711.2*, *XM_015773840.2*, and *XM_015776632.2* for subcellular localization assays. Just as the results showed, these three genes were all located in the nucleus, which was consistent with our previous predictions (Figure 2).

### 2.2. Phylogenetic, Motif, and cis-Acting Element Analysis of the Rice LBD Gene Family

The rice LBD gene family phylogenetic tree suggested that 36 rice LBD genes could be classified into Class I and Class II, of which 31 genes belonged to Class I and 5 genes belonged to Class II. Since the protein sequences of *XM_026021321.1* and *XM_015762595.2*, which were classified as Class I, were identical, they showed no evolutionary differences in the evolutionary tree.

Then, we analyzed motifs of their amino acid sequences. A total of 13 motifs were identified, and were named as Motif 1~Motif 13, among which Motif 1, Motif 2 and Motif 3 comprise the conserved LOB domain. Motif 1 contains the highly conserved CX2CX6CX3C zinc finger structure, Motif 2 contains the LX6LX3LX6L leucine zipper structure, and Motif 3 contains the GAS (Gly-Ala-Ser). All genes except *XM_015794036.2* contain these 3 motifs. Combined with the above protein sequence alignment, it was found that the deletion of Motif 1 and Motif 3 in *XM_015794036.2* might be caused by a change in one of its bases during its evolution. In addition to the 3 conserved motifs, many members also contain other, different motifs that may have been formed during evolution. As core functional elements, motif evolution generates novel functions, enhances environmental adaptation, promotes domain recombination, and maintains plasticity. These processes make motifs a key molecular driver of species diversity and adaptive evolution. Therefore, differences in motifs among LBD gene family members likely underlie their functional divergence.(Figure 3).

The *cis*-acting element analysis revealed many cis-acting elements with different functions in the LBD gene family of rice (Figure 4). In addition to the basic *cis*-acting elements, there are mainly four types of elements: light-responsive elements, plant growth and development-responsive elements, phytohormone-responsive elements, and stress-responsive elements. The analysis results of *cis*-acting elements revealed the functional diversity of rice LBD gene family members, which participate in various processes of plant growth and development. Among them, *XM_015769048* has a large number of *cis*-acting elements, which may be multifunctional, while X*M_015774606* has a small number of *cis*-acting elements, which are mainly related to hormones and stress, and may be involved in the response to drought stress.

In general, LBD gene family members have significantly fewer plant growth and development response elements than the other three types; light response elements are mainly concentrated on Box4 and G-Box, which play crucial roles in leaf development and senescence in rice [39,40], and the involvements of LBD genes containing Box4 and G-box in promoter in regulation of leaf development and senescence are noteworthy; stress-responsive elements are predominantly concentrated in AREs (Anaerobic Response Elements), suggesting that members of the LBD gene family may be involved in the anaerobic induction process in rice. Additionally, the prevalence of plant hormone-related cis-acting elements suggests that LBD gene family members are likely involved in the response and regulation of abscisic acid, methyl jasmonate, and auxin.

### 2.3. Phylogeny Analysis of LBD Gene Family

The LBD gene interspecific phylogenetic tree comprises 36 rice LBD proteins, and 43 *Arabidopsis thaliana* LBD proteins (Figure 5). The results showed that proteins from the two species can be classified into two major classes, Class I and Class II. Among them, Class I has a complete leucine zipper-like motif, while Class II contains an incomplete leucine zipper-like motif. Class I can be further grouped into Class I a, Class I b, Class I c, Class I d, and Class I e subfamilies. Respectively, 31 and 5 rice LBD proteins were distributed in Class I and Class II, which is consistent with the results of the intraspecific phylogenetic tree.

10, 10, 9, 4 and 3 *Arabidopsis thaliana* LBD genes were clustered in 10 Class I a, 7 Class I b, 4 Class I c, 4 Class I d and 6 Class I e rice genes, respectively. Meanwhile, Class II AtLBD genes containing six genes are clustered with five class II rice genes.

### 2.4. Collinearity Analysis of LBD Gene Family in Rice

Duplication is considered to be an important means of providing the basis for evolution, and many gene family members were generated through gene duplication events during evolution [41,42]. Different gene classes tend to be retained following single and whole genome duplication, which is simultaneously related to the evolutionary direction of genes, and gene duplication often leads to gene family expansion [43]. Collinearity analysis of the LBD genes in rice and *Arabidopsis thaliana* showed that there were seven pairs of homologous genes in rice, four homologous pairs in *Arabidopsis thaliana*, and only five homologous pairs between rice and *Arabidopsis thaliana* (Figure 6). These findings reveal dual evolutionary patterns in the LBD gene family. First, the absence of extensive segmental duplications and relatively weak purifying selection pressures suggests evolutionary constraint on genomic architecture. Second, the 150-million-year divergence since the monocot-dicot split (represented by rice and Arabidopsis, respectively) has driven lineage-specific adaptation of LBD genes governing developmental processes and stress adaptation, reflecting their ecological specialization.

### 2.5. Analysis of Expression Patterns of Rice LBD Genes

The expression data of rice LBD genes were acquired by real-time fluorescence quantitative PCR from rice roots, stems, leaves and seeds (Figure 7). The results showed significant differences in the expression of LBD genes across different tissues, and could be clustered into four categories: the expression of genes in category I (RS group: roots and stems) was relatively low in leaves and seeds, with higher expression levels in both roots and stems. Category II (RL group: roots and leaves expression) genes showed relatively higher expression levels in roots and leaves, suggesting that it may be involved in the cell differentiation process. Category III (L group: leaves) was only highly expressed in leaves, which might be related to the conserved LOB domain of the LBD gene itself. The transcripts of category IV (RLD group: roots, leaves and seeds) genes accumulated in roots, leaves and seeds, while they were poorly expressed in stems.

Combined with the previous research, we found that the members of the LBD gene family in rice may have similar functions as their homologous genes in *Arabidopsis thaliana*. AtLBD16 and AtLBD29 have been demonstrated to play an important role in the formation and development of lateral roots of *Arabidopsis thaliana*. The homologous genes of these two genes in rice, *XM_015770711.2*, *XM_015776632.2*, and *XM_015792766.2*, also show high expression in rice roots. We speculated that as their *Arabidopsis thaliana* homologues, they may also contribute to the formation and development of lateral roots in rice. Furthermore, the homologous genes *XM_026021321.1* and *XM_015762595.2* share identical gene structures and exhibit consistent expression profiles, both showing high expression in leaves and seeds, which suggest potential functional redundancy between them. In contrast, another pair of homologous genes, *XM_015769048.1* and *XM_015783373.2*, while possessing identical gene structures, display completely divergent expression patterns: the former is specifically highly expressed in leaves while the latter shows root and seed preferential expression. This expression divergence correlates with significant differences in cis-regulatory element composition within their promoter regions, demonstrating the independent evolutionary trajectories of LBD homologous genes.

### 2.6. Transcriptome Data Analysis of Phytohormone and Stress Treatment

We obtained the expression data of LBD genes in rice(*Nipponbare)* roots treated with abscisic acid(ABA), jasmonic acid (JA), and cold and drought stresses, from the Rice RNA-seq Database in plantrnadb. Some LBD genes were not detected, so their expression levels in the control group and several treatment stages were marked as 0 in the figure. These data were also consistent with the results of gene expression measurements, as shown in Figure 7, which suggested that these genes were expressed at low levels in the rice roots.

Gene expression after JA treatment showed that only a small number of LBD gene family members showed a JA responsive expression pattern at different stages (Figure 8A). Some of them exhibited JA-induced expression; for example, *XM_015772758.2* and *XM_015755677.2* showed an increase in expression levels after 12 h of JA treatment, while *XM_015776632.2*, *XM_015773867.2*, and *XM_015774114.2* showed an early response to JA with increased expression after only 1 h of treatment, and *XM_015792520.2* showed a rapid decrease in expression level after 1 h of JA treatment, which suggested that they may directly or indirectly respond to JA. In *Arabidopsis thaliana*, AtLBD16 is not only a key factor in auxin-mediated lateral root formation, but also closely related to the formation of root knot nematode galls. Previous studies have found that AtLBD16 is mainly expressed in the early and middle stages of gall formation, and its expression is regulated by auxin and induced by nematode secretion; knockout of *AtLBD16* significantly reduces the infection rate of root knot nematodes in plants [44]. It has been reported that JA plays an important role in nematode and pathogen resistance in rice, especially in root knot nematode resistance [45,46,47]. The homologous genes of AtLBD16 in rice, *XM_015770711.2* and *XM_015776632.2* showed accumulated transcripts after JA treatment, and both have JA response elements, so it is speculated that they may be involved in JA-mediated resistance process in root knot nematodes, which deserves thorough exploration.

Many studies demonstrated that ABA plays an important role in salt and drought stress responses in rice [48,49]. Therefore, we also analyzed the expression pattern of LBD genes based on the transcriptome data of ABA and drought treatment (Figure 8B,D). The results showed that *XM_015758344.2*, *XM_026021321.1*, and *XM_015762595.2*, with higher expression levels after ABA treatment, were also significantly upregulated by drought treatment, which suggested that they may be involved in the drought resistance through the ABA signal pathway. ZmLBD2 and ZmLBD5 have been demonstrated to regulate the drought resistance of *maize* [50,51]. Meanwhile, ZmLBD5 also regulates drought resistance via ABA signaling, and the expression level of its homologous gene *XM_015772758.2* in rice was significantly increased after ABA and drought treatment, so we speculated that it may have a similar function to *ZmLBD5*. In addition, ABA is also involved in the regulation process of lateral root growth in rice [52]. However, whether the LBD gene family is involved in this process via ABA signaling needs further study.

Moreover, the expression levels of 22 LBD gene family members were increased after 1 h with cold stress and then decreased, while the expression levels of six other genes were significantly decreased (Figure 8C). Among the genes induced by cold stress, 16 genes were rapidly increased within 1 h after treatment, and 2 genes were significantly increased 3 h after treatment. We speculated that it may play an important role in the early stage of plant resistance to cold stress, which needs to be further verified. Current studies on cold tolerance in rice have shown that under cold stress, abscisic acid (ABA) signal transduction is negatively correlated with cytokinin (CK) signal transduction; however, the specific regulatory relationship has not been clarified [53]. Meanwhile, some researchers have found that *MaLBD5, a homolog of LBD5 in banana* and *MaJAZ1* (*Jasmonate ZIM-domain 1*) in banana fruit may have antagonistic effects on MeJA-induced cold tolerance of banana fruit, at least partially through affecting JA biosynthesis [54]. Notably, *XM_015772758.2* exhibits upregulated expression following JA treatment and cold stress exposure. Based on functional conservation observed in its homologs, we hypothesize that this gene may improve cold tolerance in rice through positive regulation of JA biosynthesis. However, the precise molecular mechanism remains to be elucidated, including whether it directly activates JA synthase genes, modulates JA signaling components, or coordinates crosstalk with other hormonal pathways, all requiring further experimental validation.

### 2.7. Chromosomal Mapping Analysis of the LBD Gene Family in Rice

The chromosomal localization analysis results of the rice LBD gene family revealed that 36 LBD genes were unevenly distributed on 10 of the 12 rice chromosomes (Appendix A). Among them, chromosomes 4 and 6 do not contain LBD genes, while chromosome 1 has the most genes, with 12 genes, followed by chromosome 3, which contains 9 LBD genes. A total of two LBD genes are located on chromosome 2, while chromosomes 5 and 8 both exhibit four LBD genes, and chromosomes 7, 9, 10, 11 and 12 all have only one LBD gene.

Meanwhile, members with closer chromosome distance may have similar functions, for example, *XM_015782803.2*, *XM_015785013.2*, *XM_015782813.1*, *XM_015782824.1*, and *XM_026023149.1*, which are closely related on chromosome 1, are all members of Class Ie subfamily. Gene expression results showed that they all had high expression levels in rice leaves, and few in other tissues. At the same time, there are some differences in the composition of *cis*-acting elements. Similarly, *XM_015776632.2* and *XM_015776633.2*, which are closely related on chromosome 3, are members of Class Ic subfamily. Their *cis*-acting elements are highly similar, indicating that they may have common functions.

## 3. Discussion

Rice is one of the most important food crops for human beings. According to current research, the LBD gene family plays an important role in the process of growth and development as well as metabolism regulation in various plants. In this study, a total of 36 LBD genes were identified from the rice genome using rice genome-wide data from two different databases as background, which was different from previous studies [38,55] to some degree. A recent study described a reduced set of 31 members [38], also inconsistent with previous reports based on IRGSP BAC/PAC physical maps [55]—an approach now recognized as methodologically constrained due to its dependence on superseded genome assemblies. However, all 36 genes were identified as members of the LBD gene family in our study. Indeed, the differences observed in the amino acid sequences of *XM_026025700.1*, *XM_015769048.1*, *XM_015780552.2*, *XM_026027463.1*, and *XM_015782803.2* compared to previous studies, while retaining high similarity in their domains, which could be attributed to the continuous updating and refinement of rice gene databases.

Further analysis revealed an uneven distribution of the 36 LBD genes across 10 out of 12 rice chromosomes, with potential functional similarities among closely clustered members. While exhibiting variations in physicochemical properties, all identified LBD proteins were predicted to localize to the nucleus. These findings diverge from prior reports in two key aspects: (i) the absence of LBD gene annotations on chromosomes 1–3 in our current dataset, and (ii) the nuclear-specific localization pattern, contrasting with earlier predictions of mitochondrial, chloroplast thylakoid membrane, or extracellular localization for certain family members. Therefore, we selected three genes (XM_015770711.2, XM_015773840.2, and XM_015776632.2) with different predicted subcellular localization results for experimental validation. The detection results showed that the expression positions of the three genes were all in the nucleus, which was in line with our previous prediction results (Figure 2).

We conducted a phylogenetic tree analysis of the LBD gene family in rice and *Arabidopsis thaliana*. Our findings revealed that 31 rice LBD genes and 37 *Arabidopsis thaliana* genes clustered into Classes I a–I e, whereas 5 rice genes and 6 *Arabidopsis thaliana* genes grouped into Class II. This is consistent with the results of the developmental evolutionary tree among species of LBD genes, and the genes clustered into the same class likely share similar functions; thus, the above results provide a certain basis for predicting the functions of rice LBD genes. Previous studies identified a distinct ‘At Class’ in *Arabidopsis thaliana* consisting of 24 genes (AtLBD1-12, AtLBD21-28, AtLBD32, AtLBD33, AtLBD35, AtLBD36), which did not cluster with rice genes [55]. However, in our study, these genes clustered with Class I a, Class I c, and Class I d rice genes, suggesting that differences in the sequence types (amino acid vs. nucleotide) used for constructing phylogenetic trees may contribute to these discrepancies. This discrepancy might stem from the utilization of nucleotide sequences in the previous phylogenetic tree construction. Actually, amino acid sequences are generally more susceptible to convergent evolution than nucleotide sequences, which can lead to discrepancies between phylogenetic trees constructed using amino acids versus nucleotides. The choice of sequence type critically impacts phylogenetic reconstruction accuracy. Nucleotide sequence alignment presents three key limitations: (i) requirement for precise reading frame maintenance, (ii) technical challenges in non-coding region (e.g., intron) alignment that may introduce artifacts, and (iii) susceptibility to synonymous mutation noise. In contrast, amino acid sequence alignment offers distinct advantages: (i) enhanced conservation of functional protein domains, (ii) automatic filtering of codon degeneracy effects, and (iii) reduced impact of synonymous substitutions on tree topology. Consequently, our phylogenetic analysis employed amino acid sequences to maximize evolutionary signal detection while minimizing systematic errors. In addition, recent phylogenetic analyses by using amino acid sequence involving the *Arabidopsis thaliana* LBD gene family and other species have not uncovered the existence of such an AT class [7,8,9,10].

Meanwhile, this study comprehensively analyzed the physical and chemical properties, motifs, cis-acting elements, collinearity, and gene expression of LBD gene family in rice. There are some differences in the physical and chemical properties between members of LBD gene family (Table 1), and the motif analysis results show that except for the three motifs which constitute the LOB domain, the other motifs are different, which preliminarily demonstrates the possible functional differences among family members; the collinearity analysis results show that only seven pairs of genes from the LBD gene family appear in rice, indicating that they may not have experienced large-scale fragment replication events and strong purification selection in the evolutionary process. The data of *cis*-acting elements (Figure 4) and transcriptome of phytohormones treatment (Figure 8) show that LBD gene family members are involved in the response and regulation process of various phytohormones, indicating that LBD gene family members may directly or indirectly participate in various aspects of rice growth and development through the response and action mechanism of various phytohormones in rice, which has been confirmed by other studies. IAA13- and ARF19-dependent signaling pathways stimulate the transcription of OsLBD1-8 (XM_015780552.2) to enable auxin to act, including the formation of constitutive aerenchyma (CA) and lateral roots (LR) [22]. Meanwhile, a previous study has demonstrated that its homologous gene LBD13 is expressed in emerged LRs and LR meristems and regulates LR formation in *Arabidopsis thaliana* [56], which was consistent with our analysis of the results that there are many auxin-responsive elements on its promoter. Overexpression of these genes may promote lateral root proliferation and enhance adaptability in arid or infertile soils. In addition, LBD12-1 transcription factor inhibited the size of the apical meristem by inhibiting the expression of *AGO10* (*Argonaute 10*) [57]. Its promoter region had two meristem expression response elements CAT-box. Repression of *OsIG1(XM_015779215.2)* expression leads to the appearance of unusual double ovules and various abnormalities in the development of floral organs and megasporophylls in rice. It also participates in the regulation of the division, differentiation, and lateral growth of spherical cells during leaf development [29]. Cis-regulatory element analysis showed that the promoter of OsIG1 contains numerous hormone-related response elements, including two endosperm-specific negative expression elements, suggesting that OsIG1 may regulate rice organ development through a phytohormone-dependent pathway. Inhibiting the expression of OsIG1 can explore the application of the double-ovule phenotype in hybrid breeding, or improve seeds setting rate through hormone (such as gibberellin) regulation. Crown Rootless (CRL1)/XM_015776633.2 plays a crucial role in the formation of rice crown roots, likely through the ARF signaling pathway [58]. Meanwhile, the promoter of *CRL1* contains four JA response elements. JA can promote crown roots growth and development in rice [59], which suggests that CRL may be a key node in the regulation of crown root formation by auxin and jasminate. A recent study found that *OsLBD16* is a direct downstream target of WUSCHEL-related homeobox(WOX11), while overexpressing *OsLBD16* in *wox11*-deficient plants can partially restore the root crown deficient phenotype [60]. Previous studies have also shown that WOX11 and CRL1 cooperate to promote the development of rice crown roots [61]. The phylogenetic tree shows that *OsCRL1 (XM_015776633.2/XM_015792766.2)* is closely related to *OsLBD16*(*XM_015770711.2/ XM_015776632.2*), both of which play important roles in the regulation of rice crown root development. Enhancing root structure through gene editing (such as CRISPR/Cas9-mediated overexpression) or promoter engineering can improve the lodging resistance and nutrient absorption efficiency of rice. Overexpression of *OsLBD3-7*(*XM_026023997.1*) reduces the size and number of blister-like cells, resulting in narrower leaves and axial curvature [62], which is consistent with its high expression in rice leaves. Targeted editing of this gene may optimize the plant architecture and improve the photosynthetic efficiency of the population.

The members of the class II family with higher conservation may also have similar functions, such as *XM_015791721.2* and *XM_015773867.2*, which are close to each other in phylogenetic trees, and the expression patterns are basically same after cold and drought treatments. The expression patterns of XM_15788984.2 and XM_15772758.2 (both Class II), as well as XM_126021321.1 and XM_15762595.2 (both Class Ia), were similar under these treatments, suggesting potential functional redundancy among these genes. The members of the LBD gene family that are closely related on the phylogenetic tree likely share similar functions, but experimental validation is needed to confirm this hypothesis. When investigating these genes’ functions, null mutants or gene-silenced lines may exhibit no phenotypic changes under normal conditions but display distinct phenotypes under specific stresses. For instance, OsLBD12-1 knockout plants show wild-type morphology under control conditions but develop enlarged shoot apical meristem under salt stress [57]. Furthermore, the absence of phenotypic changes in mutants may result from functional redundancy among homologous genes, necessitating the generation of multiple-gene knockout lines for comprehensive functional analysis.

Combined with the existing research and transcriptome data, we hypothesize that the members of the LBD gene family in rice may have similar functions as their homologous genes in other plants. For example, *XM_015770711.2* and *XM_015776632.2* may be involved in the resistance process of root knot nematodes and the formation and development of lateral roots of plants, like *AtLBD16*; *XM_015772758.2* may also affect the cold resistance of rice by regulating the biosynthesis of JA, similar to its homologous gene *MaLBD5* in banana. Meanwhile, the homologous gene *ZmLBD5* of *XM_015772758.2* in maize may have the same function; by regulating the drought tolerance of plants through ABA, it can serve as a target for drought resistance genetic engineering to improve the drought tolerance of rice. Therefore, through the exploration and analysis of other homologous genes and transcriptome data, we found that the functions of some genes in rice may be the same as their homologous genes in other plants, especially the more conservative class II family members, The high conservation and expression consistency of Class II genes (such as *XM_015791721.2* and *XM_015773867.2*) suggest that their functional redundancy may provide a “backup” mechanism for stress resistance. Through gene pyramiding or synergistic regulation, the stability of crops under multiple stresses can be enhanced. This also lays a certain foundation for the subsequent in-depth exploration of the specific functions of the LBD gene family and provides some assistance for its application in rice agricultural production.

Naturally, we must acknowledge the limitations of our study. On one hand, while we conducted a preliminary analysis of the evolutionary relationships between LBD gene families in rice and Arabidopsis thaliana, which revealed the significant influence of ecological environments on the evolution of homologous genes across species, we lacked an in-depth investigation into the evolutionary relationships of LBD genes among different rice varieties. On the other hand, our understanding of the functions of rice LBD genes is entirely based on bioinformatic analyses and functional studies of homologous genes, without direct experimental validation.

In future studies, a more comprehensive analysis of the evolutionary divergence of the LBD gene family across various rice cultivars will help us better understand the adaptive changes in individual members throughout hundreds of millions of years of rice evolution, as well as their key roles in rice growth and development. Undoubtedly, all bioinformatic predictions must be supported by solid experimental evidence. A full understanding of the functional roles of LBD gene family members depends on the generation of corresponding genetic materials. For example, creating overexpression or loss-of-function mutants of XM_015770711.2, XM_015776632.2, and XM_015792766.2 will enable us to investigate their specific roles in rice root development and evaluate their potential applications in improving drought and salt tolerance.

## 4. Materials and Methods

### 4.1. Sources of Data

In this study, the rice genome and annotation files come from Rice Genome Annotation Project (http://rice.uga.edu/pub/data/Eukaryotic_Projects/o_sativa/annotation_dbs/pseudomolecules/version_7.0 (accessed on 18 May 2023)) [63] and NCBI (https://www.ncbi.nlm.nih.gov/datasets/genome/GCF_001433935.1/ (accessed on 18 May 2023)). We obtained the protein sequences of *Arabidopsis thaliana* gene family members from the Plant Transcription Factor Database (https://planttfdb.gao-lab.org/ (accessed on 27 May 2023)). The data for the conserved LBD domain (LOB domain, DUF260; pfam number: pfam03195) were acquired from Pfam (https://pfam.xfam.org/ (accessed on 18 May 2023)) [64].

### 4.2. Identification and Physicochemical Properties Analysis

In this study, we screened the rice LBD gene family members by two methods: ① We utilized HMMER3 (http://hmmer.org/ (accessed on 19 May 2023)) to construct the Hidden Markov (HMM) model of the LBD gene family, and through this model and the conserved domains of LBD genes, we screened out the genes that might be members of the LBD gene family (E-value ≥ 1 × 10^−10^) [65]. ② By BLASTP alignment, the obtained protein sequences of *Arabidopsis thaliana* LBD gene family members were used to obtain the target sequences by comparing with the rice protein database. Then, we compared the sequences obtained by the two methods, the two sets of results were found to be the same.

We verified in domains by Batch CD-Search (https://www.ncbi.nlm.nih.gov (accessed on 28 May 2023)) [66] and SMART (https://smart.embl.de/ (accessed on 28 May 2023)), and used ClustalW for multiple alignment. Physicochemical property identification and the analysis of protein sequences were carried out by the ExPASy (https://www.expasy.org/ (accessed on 29 May 2023)) online protein analysis website [67], and subcellular localization prediction of the identified LBD genes was performed using Cell-PLoc 2.0 (http://www.csbio.sjtu.edu.cn/bioinf/Cell-PLoc-2/ (accessed on 29 May 2023)) [68].

### 4.3. Subcellular Localization

The CDS sequences of *XM_015770711.2*, *XM_015773840.2* and *XM_015776632.2* genes were constructed into pCAMBIA1300-35S-GFP vector and then transfected into leaves of tobacco (Each gene is inoculated on 3–4 leaves.). After 48 h of culture, the samples were collected and stained with DAPI, and finally observed by confocal microscope (LSM710, ZEISS, Oberkochen, Germany).

### 4.4. Chromosomal Mapping Analysis

Chromosomal distribution analysis of LBD gene family members was carried out by Tbtools (https://github.com/CJ-Chen/TBtools-II (accessed on 9 June 2023)) [69], and the screened LBD gene information and the rice genome gff annotation files were imported into TBtools (Version 2.112) to obtain the chromosomal location and gene structure information of the rice LBD gene family members.

### 4.5. Analysis of Motifs, Gene Structures, and cis-Acting Elements

Multiple sequence alignment of rice LBD protein sequences was carried out by ClustalW [70], and we utilized the results of multiple sequence alignment to construct a phylogenetic tree of rice LBD genes by MEGA X (https://www.megasoftware.net/dload_win_gui (accessed on 12 June 2023)) [71], using the maximum likelihood method (ML, Maximum Likelihood) with the JTT Model and the calibration parameter Bootstrap set to 1000 times. The online tool MEME (https://meme-suite.org/meme/tools/meme (accessed on 14 June 2023)) [72] was introduced to analyze the motif of rice LBD proteins. The 1500 bp promoter region upstream of the rice LBD gene was analyzed for *cis*-acting elements by the online tool PlantCARE (https://bioinformatics.psb.ugent.be/webtools/plantcare/html/ (accessed on 15 June 2023)) [73] and visualized using TBtools.

### 4.6. Phylogeny Analysis

Multiple sequence alignment of the LBD protein sequences of rice and *Arabidopsis thaliana* was carried out by MUSCLE, and the results of multiple sequence alignment were used in MEGA X software (Version 10.2.5) using the Neighbor-Joining method (NJ. Neighbor-Joining) to construct a phylogenetic tree of the LBD gene family, using JTT (Jones–Taylor–Thornton) Model with the calibration parameter Bootstrap set to 1000 times, while ITOL (https://itol.embl.de/ (accessed on 18 June 2023)) [74] and AI (Adobe Illustrator) were used for visualization.

### 4.7. Collinearity Analysis

TBtools was used to analyze and visualize the collinearity of rice and *Arabidopsis thaliana* LBD gene family members.

### 4.8. Expression Patterns Analysis

Rice material was selected from rice at about 10 days after filling, and its roots, stems, leaves and seeds parts were taken. Samples were frozen in liquid nitrogen, stored at −80 °C, and later ground for RNA extraction. Then, reverse transcription was used to obtain cDNA, and, finally, real-time fluorescence quantitative PCR (Cyclic number 40) was carried out to acquire rice LBD gene expression data, which was visualized using TBtools.

### 4.9. Analysis of Transcriptome Data of Phytohormone and Stress Treatment

Transcriptome data of LBD gene family members in the root of rice at different times after ABA, JA, cold, and drought treatments were obtained and screened from the Rice RNA-seq Database in the online website plantrnadb (https://plantrnadb.com/ricerna/ (accessed on 17 January 2024)), and were visualized by using TBtools.

## 5. Conclusions

As the rice genome data have been updated, we see that some previous studies analyzing the LBD gene family in rice show inconsistencies with the current genome data, which is particularly reflected in the number of family members (such as the 36 newly identified members) and differences in physicochemical properties (amino acid sequence lengths of 103–456, nuclear localization prediction) and the interspecific phylogenetic tree [38,55]. This may be a data conflict caused by the iteration of the genome version. Therefore, we re-screened and identified the LBD gene family in rice using two different databases (Rice Genome Annotation Project and NCBI). It was finally confirmed that the rice LBD gene family contains 36 members, and their functional characteristics were systematically analyzed. These findings not only resolve the data conflicts caused by the iterative versions of the genome, providing a standardized benchmark for subsequent functional studies, but also offer potential targets for molecular design breeding by clarifying the hormone regulatory network and root development mechanism. As well as reported LBD genes such as OsLBD1-8 and CRL, *XM_015770711.2*, *XM_015776632.2* and *XM_015792766.2* also can be applied as potential targets for optimizing root architecture. Gene expression profiling revealed that XM_015770711.2, XM_015776632.2, and XM_015792766.2 exhibit root-specific expression patterns. Their Arabidopsis homologs, AtLBD16 and AtLBD29, have been functionally demonstrated to regulate lateral root formation and development, suggesting that these rice LBD genes may similarly control root morphogenesis in rice. Furthermore, XM_015772758.2 represents a potential key target for improving environmental adaptation in rice. This gene shows significant upregulation under ABA treatment and drought stress, and its maize ortholog, ZmLBD5, has been reported to enhance drought tolerance via ABA signaling. Additionally, XM_015772758.2 is also induced by JA and cold stress, indicating its crucial role in rice–environment interactions and its high potential for agricultural applications, such as developing drought- and cold-resistant rice varieties. This research lays a theoretical foundation for the targeted improvement of rice root structure, stress resistance and yield traits by regulating the LBD gene regulatory network, and has potential practical value in promoting precise breeding strategies and addressing global food security challenges.

## Figures and Tables

**Figure 1 ijms-26-03948-f001:**
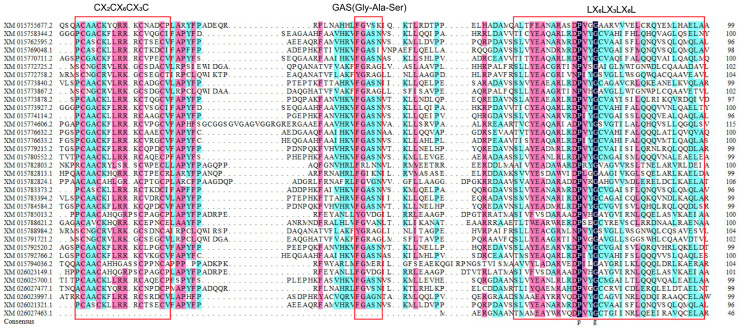
Rice LBD family conserved domain protein sequence alignment.CX_2_CX_6_CX_3_C, Gly-Ala-Ser and LX_6_LX_3_LX_6_L constitute the LOB domain.

**Figure 2 ijms-26-03948-f002:**
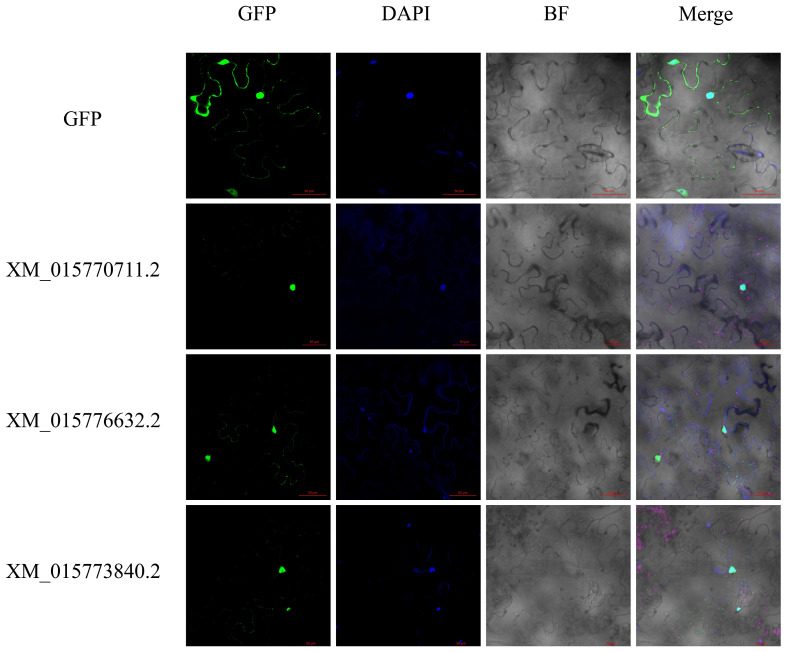
Subcellular localization of *XM_015770711.2*, *XM_015773840.2*, and *XM_015776632.2*. GFP is green fluorescent protein, DAPI is DAPI staining, BF is bright field, and Merge is a superimposed graph.

**Figure 3 ijms-26-03948-f003:**
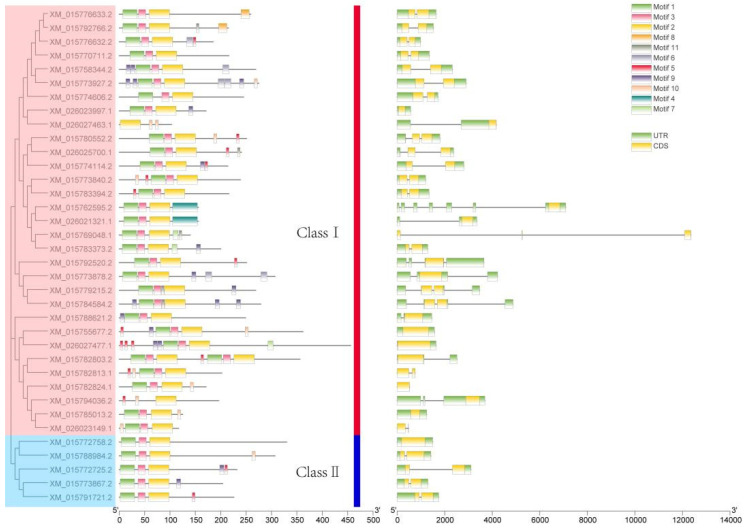
Analysis of phylogenetic tree, motif-structure, and gene structure of *LBD* gene family in rice.

**Figure 4 ijms-26-03948-f004:**
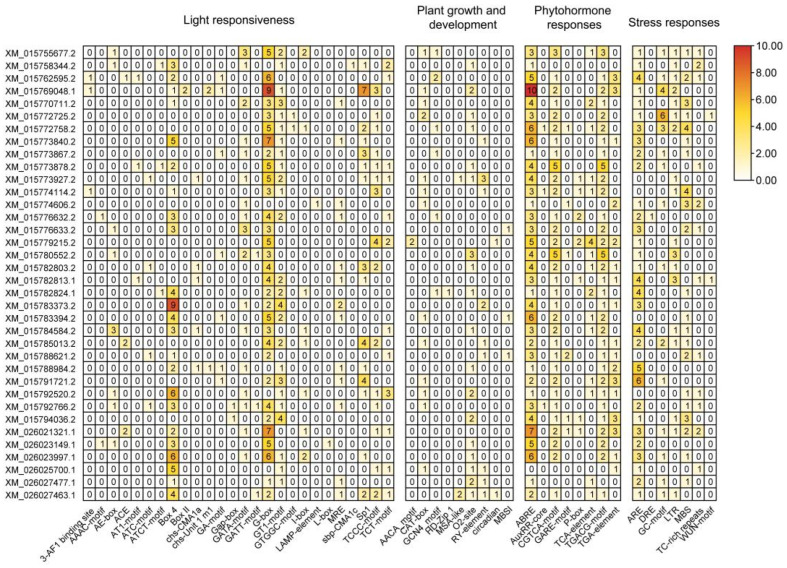
Identification of *cis*-acting element analysis in the upstream 2000 bp promoter region of the rice *LBD* gene. The figures in the figures are the number of *cis*-acting elements that the gene has.

**Figure 5 ijms-26-03948-f005:**
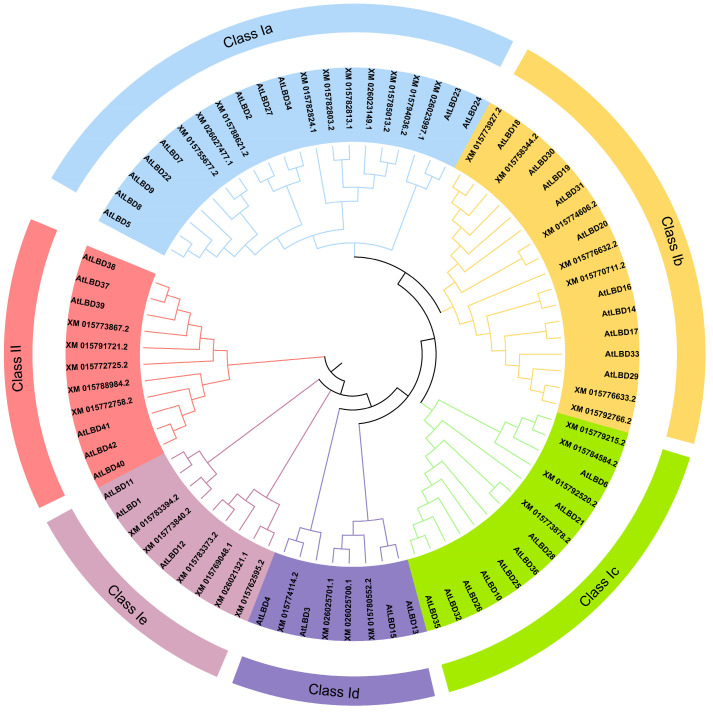
*LBD* gene interspecific phylogenetic tree. All genes can be divided into 2 subclasses, among which Class I has 5 subclasses.

**Figure 6 ijms-26-03948-f006:**
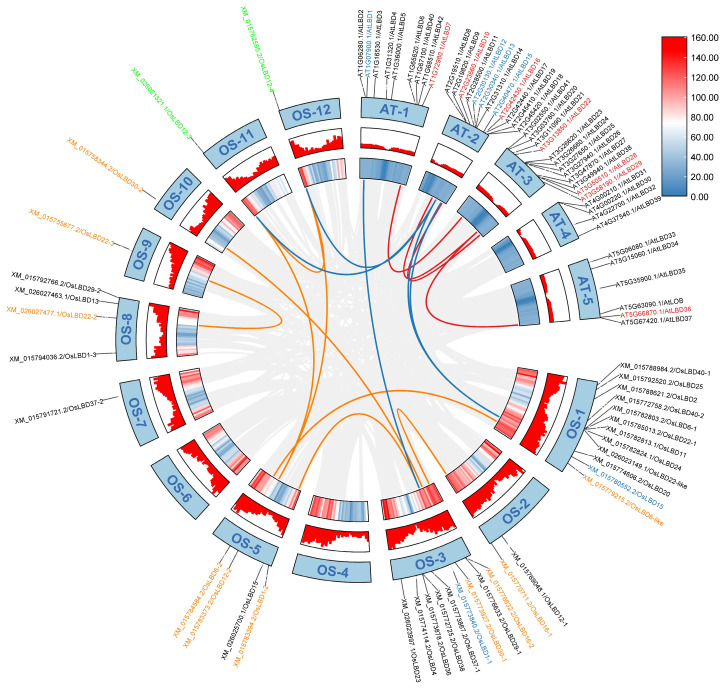
Collinearity analysis of *LBD* genes in rice and *Arabidopsis thaliana.* Orange: collinear relationships within rice; red: collinear relationships within *Arabidopsis thaliana*; blue: colinear relationship between rice and *Arabidopsis thaliana*; green: collinear relationships within rice as well as between rice and *Arabidopsis thaliana*.

**Figure 7 ijms-26-03948-f007:**
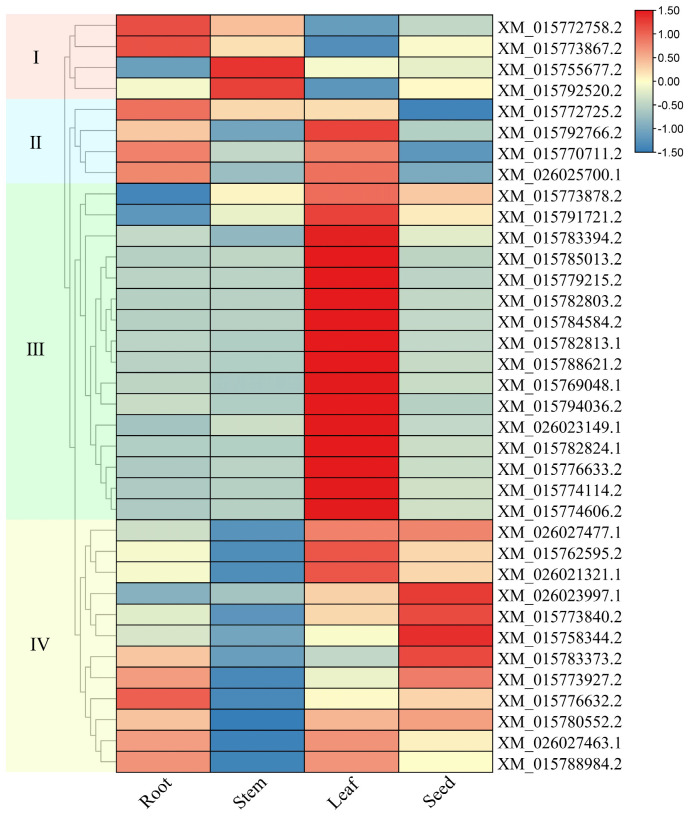
The gene expression heatmap of LBD gene family members in 4 different tissues of roots, stems, leaves, and seeds of rice. Red indicates high expression, and blue indicates low expression.

**Figure 8 ijms-26-03948-f008:**
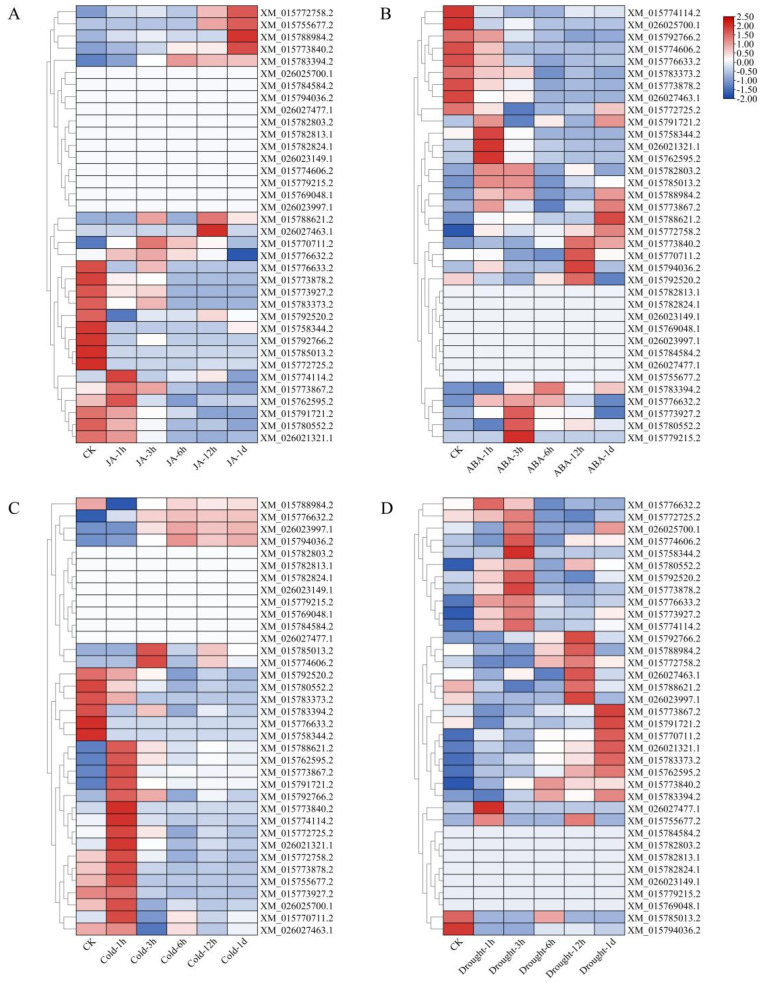
Transcriptome analysis of genes involved in phytohormone signaling and stress response. (**A**) Transcriptome data of rice at various periods and control samples after JA treatment; (**B**) transcriptome data of rice at various periods after ABA treatment and control samples; (**C**) transcriptome data of rice at various periods after cold treatment and control samples; (**D**) transcriptome data of rice at various periods after drought treatment and control samples. Red indicates high expression, and blue indicates low expression.

**Table 1 ijms-26-03948-t001:** Physicochemical properties of proteins of the LBD gene family in rice.

Accession Number	Gene ID	Gene Name	Protein Size (aa)	MV	PI	GRAVI	Subcellular Localizatio
*XM_015788984.2*	*LOC_Os01g03890.1*	*OsLBD40-1*	307	32,379.55	6.89	−0.297	Nucleus.
*XM_015792520.2*	*LOC_Os01g07480.1*	*OsLBD25/OsAS2*	251	25,728.12	7.61	0.003	Nucleus.
*XM_015788621.2*	*LOC_Os01g14030.1*	*OsLBD2*	249	26,091.37	9.72	−0.321	Nucleus.
*XM_015772758.2*	*LOC_Os01g32770.1*	*OsLBD40-2*	330	35,019.79	5.36	−0.399	Nucleus.
*XM_015782803.2*	*LOC_Os01g39040.1*	*OsLBD6-1*	356	38,510.31	4.9	−0.396	Nucleus.
*XM_015785013.2*	*LOC_Os01g39070.1*	*OsLBD22-1*	125	13,306.08	6.26	−0.425	Nucleus.
*XM_015782813.1*	*LOC_Os01g39150.1*	*OsLBD11*	202	21,537.04	7.77	−0.682	Nucleus.
*XM_015782824.1*	*LOC_Os01g39160.1*	*OsLBD24*	171	17,865.16	5.25	−0.127	Nucleus.
*XM_026023149.1*	*LOC_Os01g39220.1*	*OsLBD22-like*	117	12,500.28	6.89	−0.173	Nucleus.
*XM_015774606.2*	*LOC_Os01g56530.1*	*OsLBD20*	245	25,730.04	7.72	−0.147	Nucleus.
*XM_015780552.2*	*LOC_Os01g60960.1*	*OsLBD15-1/OsLBD1-8*	251	26,493	6.45	−0.173	Nucleus.
*XM_015779215.2*	*LOC_Os01g66590.1*	*OsLBD6-like*	269	27,621.93	8.17	−0.08	Nucleus.
*XM_015769048.1*	*LOC_Os02g48270.1*	*OsLBD12-1*	140	15,868.16	6.81	−0.481	Nucleus.
*XM_015770711.2*	*LOC_Os02g57490.1*	*OsLBD16-1*	216	22,457.64	8.09	0.103	Nucleus.
*XM_015776632.2*	*LOC_Os03g05500.1*	*OsLBD16-2*	185	19,631.01	6.05	−0.186	Nucleus.
*XM_015776633.2*	*LOC_Os03g05510.1*	*OsLBD29-1/OsCRL1-like*	259	26,736.57	5.96	−0.085	Nucleus.
*XM_015773927.2*	*LOC_Os03g14270.1*	*OsLBD30-1*	275	28,252.75	8.27	−0.321	Nucleus.
*XM_015773840.2*	*LOC_Os03g17810.1*	*OsLBD1-1*	239	25,556.79	6.06	−0.241	Nucleus.
*XM_015773867.2*	*LOC_Os03g33090.1*	*OsLBD37-1*	204	21,037.1	8.41	0.042	Nucleus.
*XM_015772725.2*	*LOC_Os03g41330.1*	*OsLBD38*	232	23,618.55	6.05	0.023	Nucleus.
*XM_015773878.2*	*LOC_Os03g41600.1*	*OsLBD36*	307	33,077.8	6.65	−0.577	Nucleus.
*XM_015774114.2*	*LOC_Os03g45750.1*	*OsLBD4*	214	22,012.96	9.06	−0.131	Nucleus.
*XM_026023997.1*	*LOC_Os03g57670.1*	*OsLBD23/OsIAL1-like*	171	18,322.4	7.63	−0.458	Nucleus.
*XM_015783394.2*	*LOC_Os05g03160.1*	*OsLBD1-2*	216	22,365.67	6.57	0.125	Nucleus.
*XM_026025700.1*	*LOC_Os05g07270.1*	*OsLBD15-2*	241	25,300.6	8.77	−0.078	Nucleus.
*XM_015783373.2*	*LOC_Os05g27980.1*	*OsLBD12-2*	200	21,105.84	6.28	−0.145	Nucleus.
*XM_015784584.2*	*LOC_Os05g34450.1*	*OsLBD6-2*	279	28,044.35	8.23	−0.004	Nucleus.
*XM_015791721.2*	*LOC_Os07g40000.1*	*OsLBD37-2*	226	23,216.21	7.54	−0.115	Nucleus.
*XM_015794036.2*	*LOC_Os08g06659.1*	*OsLBD1-3*	196	21,283.44	4.62	−0.88	Nucleus.
*XM_026027477.1*	*LOC_Os08g31080.1*	*OsLBD22-2*	456	48,653.41	4.69	−0.474	Nucleus.
*XM_026027463.1*	*LOC_Os08g40520.1*	*OsLBD13*	103	11,163.64	4.34	−0.565	Nucleus.
*XM_015792766.2*	*LOC_Os08g44940.1*	*OsLBD29-2/OsCRL1-like*	215	23,239.17	5.85	−0.16	Nucleus.
*XM_015755677.2*	*LOC_Os09g19950.1*	*OsLBD22-3/OsIAL1*	362	37,637.55	5.27	−0.397	Nucleus.
*XM_015758344.2*	*LOC_Os10g07510.1*	*OsLBD30-2*	269	27,974.12	6.41	−0.336	Nucleus.
*XM_026021321.1*	*LOC_Os11g01550.1*	*OsLBD12-3*	156	17,404.87	6.42	−0.254	Nucleus.
*XM_015762595.2*	*LOC_Os12g01550.1*	*OsLBD12-4*	156	17,404.87	6.42	−0.254	Nucleus.

## Data Availability

Data is contained within the article (and Appendix A).

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
