# Peer review of "Comprehensive Characterization and Functional Analysis of the Lateral Organ Boundaries Domain Gene Family in Rice: Evolution, Expression, and Stress Response"

_ijms, 2025, doi:10.3390/ijms26093948_

Round 1

Reviewer 1 Report

Comments and Suggestions for Authors

Dear Editor

Thanks for giving chance to review “Comprehensive Characterization and Functional Analysis of the LBD Gene Family in Rice: Evolution, Expression, and Stress Response” for publication in International Journal of Molecular Science. The study is comprehensive and well-structured however is relying heavily on bioinformatics with lack of experimental validation. While the study identifies LBD genes and their expression patterns, it provides limited insight into their biological functions and regulatory mechanisms. Likewise, uneven distribution of LBD genes across chromosomes may not necessarily reflect evolutionary or functional significance. The study identifies 36 LBD genes but does not fully characterize their physiological roles or regulatory networks. Finally, study does not fully explore the broader implications of its findings for crop improvement or evolutionary biology.

Please see attached pdf for specific examples and suggestions

Comments on the Quality of English Language

There are several grammatical issues, technical ambiguities, and areas where the text could be improved for clarity and precision.

Reviewer 2 Report

Comments and Suggestions for Authors

Dear Authors,

I have read your manuscript and found it interesting. However, the following points should be considered to improve its clarity and impact:

1.        Abstract: Break long sentences, reduce parentheses, and avoid repetitive phrases. Inconsistent naming of the LBD gene family (e.g., “LBD (lateral organ boundaries domain)” vs. “LBD (LOB domain-containing)”). Keep it consistent. Moreover, please separate physicochemical properties from functional analysis and better structure expression profiling findings. Also, clearly highlight the study’s significance for rice breeding and stress resistance.

2.        Introduction: Improve transitions between sections, especially when shifting from gene function to rice-specific research. Moreover, please clearly state the research gap and emphasize the significance of analyzing LBD genes in rice.

3.        Results,

·      section 2.1: In the discussion, I want you to briefly explain why BLASTP and HMMER were chosen and why the Rice Genome Annotation Project was revisited.

·      Section 2.2: Standardize the use of “motif” and “cis-acting elements” throughout. Also, clarify the evolutionary importance of motif variations.

·      Section 2.4: Highlight the significance of fewer homologous pairs between rice and Arabidopsis thaliana.

·      Section 2.5: Briefly define what “Category I,” “Category II,” etc., represent for easier understanding. Furthermore, Ensure consistency in how “roots,” “leaves,” and “seeds” are referred to throughout.

·      Section 2.6: Emphasize the preliminary nature of speculations, especially regarding JA and cold tolerance.

4.        Discussion: Provide clearer context on how this study’s findings differ from previous research, especially regarding the LBD gene sequences and distribution across rice chromosomes. Moreover, Clarify the rationale behind the discrepancies observed between nucleotide and amino acid-based phylogenetic trees and their potential impacts on the results. Propose more specific functional experiments or gene knockout studies to confirm the roles of closely related LBD genes in rice root and shoot development. Explore the potential implications of functional redundancy among genes within the same class and its impact on plant growth and development. Discuss the potential applications of your findings in agricultural biotechnology, especially regarding stress resistance, root development, and crop improvement.

5.        Materials and methods:

·      Provide more details on the versions of software and tools used (e.g., HMMER3, MEGA X, TBtools) to ensure reproducibility.

·      Clarify the criteria for sequence filtering more explicitly, such as the parameters used in HMMER3 (e.g., E-value threshold).

·      Mention the number of replicates or controls used in the tobacco leaf transfection experiment for better experimental rigor.

·      Include more details on the real-time PCR conditions (e.g., primers, cycling conditions) to enhance reproducibility.

·      Specify the visualization tools or settings used for the data (e.g., TBtools, MEME, PlantCARE) to improve clarity.

·      If applicable, mention any statistical tests used to validate the experimental results or data interpretations.

6.        Conclusion:

·      The sentence “some previous studies analyzing the LBD gene family in rice show inconsistencies with the current genome data” could be made more specific. Which studies are being referred to, and how exactly do they show inconsistencies?

·      The conclusion could benefit from more logical progression. For example, discuss the findings first, followed by their implications, and conclude with a clear statement about future directions.

·      The conclusion could emphasize the broader significance of the findings, such as how understanding LBD genes could contribute to rice breeding or agricultural advancements.

·      The sentence “LBD gene family members are functionally diverse” could be merged with earlier statements to avoid redundancy.

Round 2

Reviewer 2 Report

Comments and Suggestions for Authors

Dear Authors,

Thank you for submitting the revised version of your manuscript. I have carefully reviewed the updated version and acknowledge that it demonstrates significant improvements. Below are my observations:

Strengths of the Revision:

  1. The overall readability of the manuscript has improved considerably, with clearer sentence structures and better paragraph transitions.
  2. The introduction is more concise and well-structured.
  3. The discussion section is more focused and better aligned with the results.
  4. The materials and methods section is now more clearly described and easier to follow.
  5. The conclusion has been appropriately revised and is now more aligned with the study's objectives and findings.

Suggestions for Further Improvement:

  1. Although the abstract has improved, I recommend including specific findings or key results to better represent the content of the manuscript. The abstract should provide a concise summary of the entire study.
  2. Please consider adding a clear and specific hypothesis at the end of the introduction to provide a solid foundation for your research objectives.
  3. The quality of Figures 1, 3, and 4 should be enhanced. In their current form, the text and graphical elements are difficult to read. High-resolution images and clearer labels would improve their impact and interpretability.
  4. I recommend including a dedicated section on Limitations to acknowledge any constraints of your study, as well as a section on Future Perspectives to suggest directions for further research.

Once these points are addressed, I believe the manuscript will be significantly strengthened.
